# Multiple Secondary Healthcare-Associated Infections Due to Carbapenem-Resistant Organisms in a Critically Ill COVID-19 Patient on Extensively Prolonged Venovenous Extracorporeal Membrane Oxygenation Support—A Case Report

**DOI:** 10.3390/microorganisms10010019

**Published:** 2021-12-23

**Authors:** Hiroaki Baba, Hajime Kanamori, Issei Seike, Ikumi Niitsuma-Sugaya, Kentaro Takei, Kengo Oshima, Yudai Iwasaki, Yuko Ogata, Hirona Nishimaki, Daisuke Konno, Takuya Shiga, Koji Saito, Koichi Tokuda, Tetsuji Aoyagi

**Affiliations:** 1Department of Infectious Diseases, Internal Medicine, Tohoku University Graduate School of Medicine, 2-1 Seiryo machi, Aoba-ku, Sendai 980-8575, Japan; kanamori@med.tohoku.ac.jp (H.K.); issei.seike.b4@tohoku.ac.jp (I.S.); sugaya.niitsuma@icloud.com (I.N.-S.); kentarou.takei.e7@tohoku.ac.jp (K.T.); koshima@med.tohoku.ac.jp (K.O.); tokuda@med.tohoku.ac.jp (K.T.); tetsujiaoyagi@med.tohoku.ac.jp (T.A.); 2Department of Intelligent Network for Infection Control, Tohoku University Graduate School of Medicine, 2-1 Seiryo machi, Aoba-ku, Sendai 980-8575, Japan; 3Division of Infection Control, Tohoku University Hospital, 1-1 Seiryo machi, Aoba-ku, Sendai 980-8574, Japan; 4Intensive Care Unit, Tohoku University Hospital, 1-1 Seiryo machi, Aoba-ku, Sendai 980-8574, Japan; yudai.i0213@gmail.com (Y.I.); yukokosegawa@gmail.com (Y.O.); hnishimaki20200401@gmail.com (H.N.); dicek0410@yahoo.co.jp (D.K.); t-shiga@med.tohoku.ac.jp (T.S.); koji.saito.e1@tohoku.ac.jp (K.S.)

**Keywords:** COVID-19, intensive care, extracorporeal membrane oxygenation, healthcare-associated infections, multi-resistant pathogens, carbapenem-resistant, case report

## Abstract

Patients with severe Coronavirus disease 2019 (COVID-19) are at high risk for secondary infection with multidrug-resistant organisms (MDROs). Secondary infections contribute to a more severe clinical course and longer intensive care unit (ICU) stays in patients with COVID-19. A man in his 60s was admitted to the ICU at a university hospital for severe COVID-19 pneumonia requiring mechanical ventilation. His respiratory condition worsened further due to persistent bacteremia caused by imipenem-non-susceptible *Klebsiella aerogenes* and he required VV-ECMO. Subsequently, he developed a catheter-related bloodstream infection (CRBSI) due to *Candida albicans*, ventilator-associated pneumonia (VAP) due to multidrug-resistant *Pseudomonas aeruginosa* (MDRP), and a perianal abscess due to carbapenem-resistant *K. aerogenes* despite infection control procedures that maximized contact precautions and the absence of MDRO contamination in the patient’s room environment. He was decannulated from VV-ECMO after a total of 72 days of ECMO support, and was eventually weaned off ventilator support and discharged from the ICU on day 138. This case highlights the challenges of preventing, diagnosing, and treating multidrug-resistant organisms and healthcare-associated infections (HAIs) in the critical care management of severe COVID-19. In addition to the stringent implementation of infection prevention measures, a high index of suspicion and a careful evaluation of HAIs are required in such patients.

## 1. Introduction

Coronavirus disease 2019 (COVID-19) has a wide spectrum of severity ranging from mild illness that does not require treatment to severe illness that requires an intensive care unit (ICU) level of care, including intensive ventilation and extracorporeal membrane oxygenation (ECMO) [1]. Patients with severe COVID-19 are at high risk for secondary infections with multidrug-resistant organisms (MDROs) [2,3]. Secondary infections contribute to a more severe clinical course and longer ICU stays in patients with COVID-19 [2]. Secondary infection is a common complication of ECMO; however, few publications have described such infection in patients with severe COVID-19 receiving ECMO. Here, we report on multiple secondary healthcare-associated infections (HAIs) due to carbapenem-resistant organisms in a patient with severe COVID-19 on prolonged venovenous extracorporeal membrane oxygenation (VV-ECMO).

## 2. Case Presentation

A man in his 60s who smokes and has type 2 diabetes presented with fever and shortness of breath, and was admitted to an isolation unit at a university hospital for severe COVID-19 pneumonia. Despite treatment with dexamethasone and remdesivir, his condition deteriorated rapidly and he required endotracheal intubation and mechanical ventilation; therefore, he was transferred to the ICU on day two of admission. He was put on airborne, droplet, and contact precautions in a single, airborne infection isolation room, with negative pressure and an anteroom. However, on day 18 he developed a fever greater than 38.0 °C and his respiratory condition worsened further. He presented severe respiratory failure with a partial pressure of oxygen (PaO_2_) of 51 mmHg on day 21 while on the following ventilator settings: fraction of inspired oxygen (FiO_2_) of 1.0, pressure support (PS) of 17 cmH_2_O, and a positive end-expiratory pressure (PEEP) of 12 cmH_2_O; and required VV-ECMO (Figure 1). Culture from blood and the tip of the central venous catheter (CVC) were positive for *Klebsiella aerogenes* (Isolate ID: KE-1) that was not susceptible to imipenem but was susceptible to piperacillin-tazobactam, meropenem, and amikacin (Table 1). Sputum cultures were positive for *Citrobacter koseri* and *Pseudomonas aeruginosa* (Table 1) but negative for *K. aerogenes*. Urine analysis showed no pyuria (<5 white blood cells per high power field), and urine culture was negative. A blood test showed elevated serum C-reactive protein (CRP) (>10.0 mg/L; reference range, <0.3 mg/L) but low procalcitonin (PCT) (0.11 ng/mL; reference range, <0.05 ng/mL). Liver and renal functions were normal. Tests for serum 1,3-β-D-glucan, the cytomegalovirus (CMV) pp65 antigenemia assay, and stool *Clostridioides difficile* toxins were negative. Whole-body enhanced computed tomography (CT) on day 18 showed diffuse ground-glass opacities in both lung fields (Figure 2) but no other abnormal findings, including venous thromboembolism, deep abscess, and sinusitis. Despite treatment with piperacillin-tazobactam 4.5 g once every 6 h (q6h) to treat the catheter-related bloodstream infection (CRBSI), the fever persisted. Blood cultures remained positive for *K. aerogenes* 3 days after the initiation of piperacillin-tazobactam treatment, despite the removal of the CVC and exchange of the ECMO oxygenator and pump. Because of the suspected bacterial colonization of the ECMO cannulas, the antimicrobial regimen was switched from piperacillin-tazobactam to a combination of short infusion (30 min) of meropenem 2 g q8h plus amikacin 5 mg/kg q12h on day 23; however, blood culture still remained positive for *K. aerogenes* on day 26, so the short infusion of meropenem was switched to an extended infusion (3 h) [4]. Subsequently, the patient’s condition improved, and blood culture became negative for *K. aerogenes* on day 32.

Although airborne precautions were lifted on day 24 after SARS-CoV-2 reverse transcription polymerase chain reaction tests of two consecutive nasopharyngeal samples taken 24 h apart were negative, single room isolation and contact precautions were continued because of the earlier detection of imipenem-non-susceptible *K. aerogenes* (Figure 1). On day 40, respiratory function was improved such that the patient could be weaned off ECMO. However, he deteriorated again because of CRBSI due to *Candida albicans* on day 38, and ventilator-associated pneumonia (VAP) due to multidrug-resistant *Pseudomonas aeruginosa* (MDRP) on day 45 (Table 1), and required a second course of ECMO due to his worsening oxygenation (PaO_2_ of 52 mmHg with FiO_2_ of 1.0, PS of 15 cmH_2_O, and a PEEP of 12 cmH_2_O) on day 47 (Figure 1).

On day 64, he suddenly developed fever, hypotension, and tachycardia. Enhanced CT revealed a perianal horseshoe abscess that was not apparent on physical examination (Figure 3). The abscess was successfully drained, and the culture showed non-carbapenemase-producing, carbapenem-resistant *K. aerogenes* (Isolate ID: KE-2) (Table 1). The patient recovered and, after a total of 72 days of ECMO support, was decannulated from VV-ECMO on day 93, although mechanical ventilation was continued. Subsequently, the patient had two more episodes of MDRP-VAP that were successfully treated with colistin plus extended infusion meropenem and aerosolized tobramycin 90 mg q8hr in combination with intravenous piperacillin-tazobactam, respectively (Figure 1). He was eventually weaned off ventilator support and discharged from the ICU on day 138. At the time of writing this paper, on day 228, the patient is undergoing rehabilitation on a general ward, whereby continuous contact precautions are being continued because MDRP was detected in sputum. Although chest CT on day 228 shows persistent interstitial changes distributed throughout the entire lung fields (Figure 2), the patient is in a good clinical condition without respiratory support, and his neurological and nutritional status is normal.

We tested the ICU room for contamination with carbapenem-resistant organisms on day 24. We collected a total of 10 samples from 10 sites, including eight high-touch surfaces and medical devices (bed rail, bedside table, vital signs monitor, infusion pump control panel, ventilator monitor/tube, venovenous extracorporeal membrane oxygenation monitor, ultrasound imaging system probe/keyboard, and computer keyboard), the sink surface/drain, and the floor surface (Appendix A). As a result, no bacterial growth was detected in all the samples.

Both *K. aerogenes* isolates KE-1 and KE-2 were confirmed to be negative for carbapenemase genes of *bla*_IMP-1_, *bla*_IMP-6_, *bla*_VIM_, *bla*_GES_, *bla*_KPC_, *bla*_NDM_, and *bla*_OXA-48_ groups (Appendix A). Whole genome sequencing (WGS)-based phylogenetic analysis revealed isolates KE-1 and KE-2 were genetically separated (Appendix A, Appendix A).

## 3. Discussion

In our case, multiple secondary HAIs occurred despite infection control procedures that maximized contact precautions and despite the absence of MDRO contamination in the patient’s room environment, which is an important reservoir of MDROs [5]. Patients with severe COVID-19 tend to require ECMO for longer than patients with conventional acute respiratory distress syndrome and are commonly treated with empiric, broad-spectrum antibiotic therapy, which puts them at high risk of acquiring MDROs [1,2]. In patients undergoing ECMO, acquisition of MDROs significantly increases the risk for subsequent VAP and CRBSI, both of which are associated with an increased risk of death [6]. Therefore, early detection and preventive strategies for HAIs and antimicrobial stewardship programs adapted to patients with severe COVID-19 receiving ECMO are needed.

Determining the cause of fever in patients with severe COVID-19 receiving ECMO is challenging because fever can be caused by many infectious conditions, including VAP, CRBSI, catheter-associated urinary tract infection, ECMO device-related infection, *Clostridioides difficile* infection, other bacterial infections (e.g., meningitis, sinusitis, biliary tract infections, and deep abscesses), non-bacterial infections (e.g., aspergillosis, CMV infection); non-infectious conditions (e.g., cerebral infarction/hemorrhage, venous thromboembolism, adrenal insufficiency, crystal arthritis, and drug fever), and COVID-19 recurrence [2,7,8,9,10]. The accurate diagnosis of HAIs is often hampered by difficulties distinguishing HAIs from other infections and non-infectious conditions. For example, ECMO frequently masks the signs and symptoms of infections [11]. Patients with severe COVID-19 usually present diffuse bilateral lung infiltrates on chest imaging that are difficult to distinguish from signs of other respiratory infections, including VAP, pulmonary aspergillosis, and CMV pneumonia [7]. Extensive airspace opacification due to lung-protective lung strategies makes interpretation of chest radiographic findings more difficult [11]. Clinical biomarkers of inflammation, including CRP and PCT, are not specific to infections, and the inflammatory response can be caused by the host response to the ECMO circuit itself [12]. A deep abscess is often hard to diagnose, particularly in patients under general anesthesia, which masks pain [13]. A perianal abscess, which is typically caused by *Enterobacterales* and anaerobic bacteria, can be fatal and requires urgent drainage [14,15], as seen in our case. In the absence of visible signs in patients with COVID-19 sepsis, particularly in those with risk factors such as male sex, smoking, and diabetes (which are also risk factors for severe COVID-19) [13,16], physicians should consider examining the anorectal region by enhanced CT or transperineal ultrasonography [17].

To the best of our knowledge, this case represents the longest reported ECMO runtime with a successful recovery [18]. Some studies showed promising outcomes of prolonged ECMO treatment (>28 days) in patients with severe COVID-19, although the data are limited [1,18]. Survival after prolonged ECMO has improved remarkably in recent years because of technological advances and prevention and management of complications [18]; however, HAIs are still one of the most difficult challenges for healthcare professionals applying ECMO. Bacterial colonization of ECMO cannulas should be considered in patients on ECMO with persistent bacteremia of unknown origin that does not improve with removal of the CVC and exchange of the oxygenator and pump [19,20]; however, changing the cannulas is highly problematic because of the high risk of potentially lethal complications and limited vascular access [21]. Furthermore, robust data are lacking on the optimal antibiotic dose for patients on ECMO [22]. Ideally, therapeutic drug monitoring (TDM) is necessary because circuit drug loss and enlarged volume of distribution have previously been reported [22,23]; however, TDM of meropenem is not yet common practice. If this method is not available, extended infusion of meropenem and combination therapy with aminoglycosides for synergistic effects are reasonable therapeutic options for patients on ECMO with refractory meropenem-susceptible, non-carbapenemase-producing-carbapenem-resistant *Enterobacterales* bacteremia [24,25].

Therapeutic options for meropenem-resistant *Enterobacterales* and MDRP infections are very limited in Japan, where the new first-line drugs for these MDROs (i.e., ceftazidime-avibactam, meropenem-vaborbactam, imipenem-cilastatin-relebactam, and ceftolozane-tazobactam) are not available at the time of writing [24,26]. Under these conditions, combination therapy (i.e., β-lactam agent plus aminoglycoside or polymyxins) remains an important therapeutic option for severe cases, even when considering adverse effects such as nephrotoxicity and ototoxicity [25]. In this context, aerosolized tobramycin is also a promising option for treatment of MDRP-VAP [27].

Although we have presented a single case report, it nevertheless highlights the significant diagnostic and therapeutic challenges of HAIs in patients with COVID-19 receiving long-term ECMO. Previous studies demonstrated a significant reduction in the incidence of exogenous HAIs due to the stringent infection prevention and control (IPC) measures implemented during the COVID-19 pandemic [28,29]. However, preventing the development of secondary endogenous infections in patients requiring prolonged multidisciplinary treatment in the ICU remains challenging, as in this case. In addition to adequate IPC measures, a high index of suspicion and careful evaluation of HAIs are required in such patients because clinical signs and symptoms, laboratory findings, and imaging features of HAIs are usually unremarkable and non-specific, and a wide range of other infectious and non-infectious conditions need to be differentiated. Infectious diseases specialists play a crucial role in preventing and managing HAIs and in addressing antimicrobial resistance by implementing antimicrobial stewardship to provide the best care for patients with severe COVID-19.

## Figures and Tables

**Figure 1 microorganisms-10-00019-f001:**
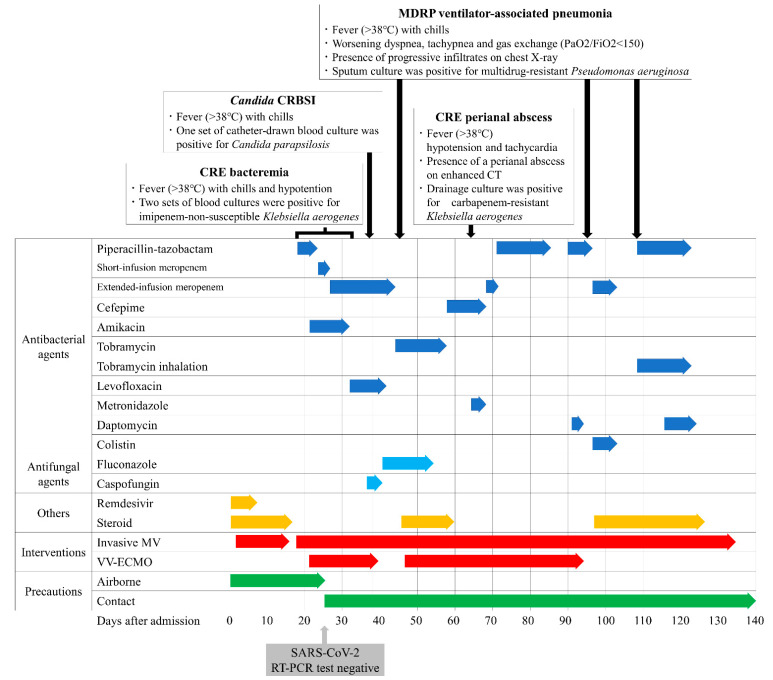
Clinical course and treatment of the present case: The patient had acquired multiple secondary infections requiring long-term broad-spectrum antibiotic use. Centers for Disease Control and Prevention criteria (https://www.cdc.gov/hai/index.html, accessed on 30 November 2021) for diagnosis of catheter-related bloodstream infection and ventilator-associated pneumonia were met. Abbreviations: CRBSI, catheter-related bloodstream infection; CRE, carbapenem-resistant *Enterobacteriaceae*; MDRP, multidrug-resistant *Pseudomonas aeruginosa*; CT, computed tomography; MV, mechanical ventilation; VV-ECMO, venovenous extracorporeal membrane oxygenation; RT-PCR, reverse transcription polymerase chain reaction.

**Figure 2 microorganisms-10-00019-f002:**
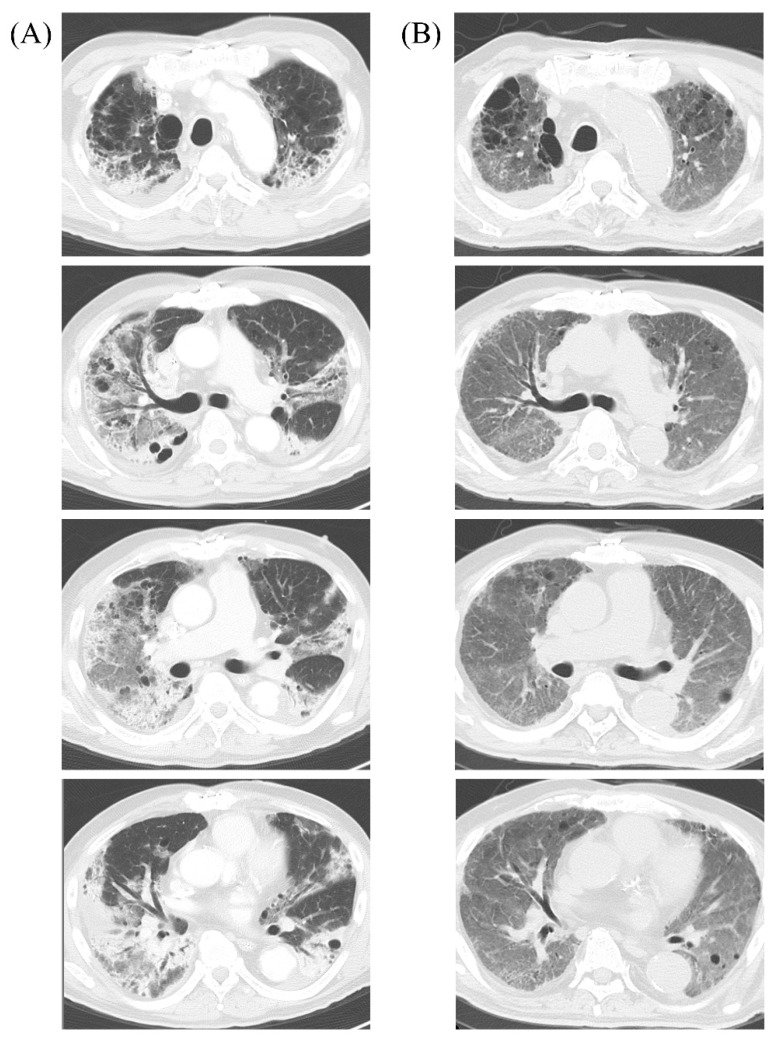
Chest computed tomography (CT) images of the present case: (**A**) CT on day 18 of admission (just before extracorporeal membrane oxygenation cannulation). Bilateral, peripheral, and basal predominant ground-glass opacities (GGOs) and consolidation were observed; (**B**) CT on day 228 (135 days post-decannulation). The CT shows significant improvement of GGOs and consolidation but persistent interstitial changes distributed throughout the entire lung fields.

**Figure 3 microorganisms-10-00019-f003:**
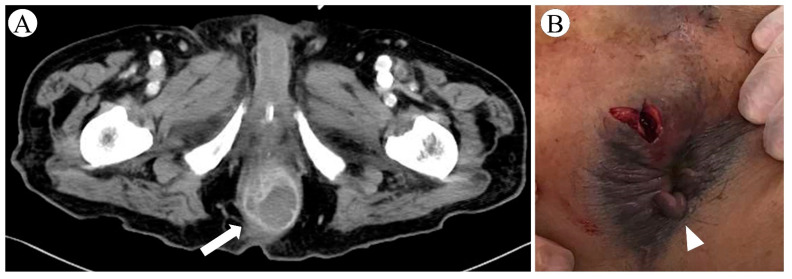
Perianal abscess in a patient with severe coronavirus disease 2019: (**A**) enhanced computed tomography revealed a horseshoe-shaped collection of fluid, 35 mm in diameter, surrounding the anus; (**B**) appearance of perianal region after incision and drainage—external examination before drainage showed only a skin tag without swelling around the anus.

**Table 1 microorganisms-10-00019-t001:** Antimicrobial susceptibility testing in detected bacterial isolates in the present case.

		MIC Value (µg/mL) and Interpretation				
Organism		*Klebsiella aerogenes* (KE-1)	*Klebsiella aerogenes* (KE-2)	*Citrobacter koseri*	*Pseudomonas aeruginosa*	MDRP
Source		Blood	Drainage Culture	Sputum	Sputum	Sputum
Antibiotics	Ampicillin	>16 R	>16 R	>16 R		
	Ampicillin-sulbactum	>16 R	>16 R	≤4 S		
	Amoxicillin-clavulanate	>16 R	>16 R	≤8 S		
	Piperacillin	16 S	64 I	>64 R	16 S	32 I
	Piperacillin-tazobactam	8 S	16 S	8 S	8 S	32 I
	Cefazolin	>16 R	>16 R	≤1 S		
	Cefmetazole	>32 R	>32 R	≤4 S		
	Ceftriaxone	≤0.5 S	>2 R	≤0.5 S		
	Ceftazidime	>8 R	>8 R	≤1 S	4 S	>16 R
	Cefepime	≤1 S	8 S	≤1 S	8 S	>16 R
	Imipenem	2 I	>2 R	≤0.5 S	1 S	>8 R
	Meropenem	≤0.25 S	>2 R	≤0.25 S	≤0.5 S	>8 R
	Aztreonam	≤1 S	>8 R	≤1 S	>16 R	>16 R
	Amikacin	≤8 S	≤8 S	≤8 S	≤4 S	32 I
	Gentamicin	≤2 S	≤2 S	≤2 S	2 S	8 I
	Tobracin				≤1 S	4 S
	Minocycline	>8 R	>8 R	4 S		
	Ciprofloxacin	<0.25 S	>2 R	<0.5 S	0.5 S	>4 R
	Levofloxacin	1 S	>4 R	≤0.12 S	2 S	>8 R
	Fosfomycin	≤4 S	>16 R	≤4 S	>16 R	>16 R
	Trimethoprim-sulfamethoxazole	≤40 S	≤40 S	≤40 S		

Bacteria and their antibiotic susceptibilities were determined by the VITEK-MS and VITEK-2 systems (Sysmex-bioMérieux Japan, Tokyo, Japan), respectively. Susceptibility results were interpreted according to available breakpoints set by Clinical and Laboratory Standards Institute (CLSI) guidelines (Performance Standards for Antimicrobial Susceptibility Testing. M100-S27. CLSI, Wayne, USA. 2018). Abbreviations: MIC, minimum inhibitory concentration; MDRP, multidrug-resistant *Pseudomonas aeruginosa*; R, resistant; I, intermediate; S, sensitive.

## Data Availability

The datasets used and/or analyzed during the current study are available from the corresponding author on reasonable request.

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
