# Peer review of "Multiple Secondary Healthcare-Associated Infections Due to Carbapenem-Resistant Organisms in a Critically Ill COVID-19 Patient on Extensively Prolonged Venovenous Extracorporeal Membrane Oxygenation Support—A Case Report"

_microorganisms, 2021, doi:10.3390/microorganisms10010019_

Round 1
Reviewer 1 Report
Baba et al. present a case of an approximately 60 years old man, who suffered from COVID-19 and was treated on ICU for 138 days. During that episode he was treated with two episodes of veno-venous ECMO and showed multiple secondary healthcare-associated infections with carbapenem-resistant Enterobacteriaceae. In the end, he could be successfully weaned off ECMO and respirator and was undergoing rehabilitation.
Merits:
This report presents a very long-lasting case of a patient with severe Coronavirus disease 2019 and shows a typical course of treatment in these patients. Therefore, this manuscript aims to give an insight in a very challenging aspect of treatment. Moreover, this case shows, that even after a long course of ICU-treatment patients still may have a positive outcome.
Limitations:
The title of this case report gives the impression, that this manuscript sets the focus on common secondary healthcare-associated infections with multidrug-resistant bacteria. Although secondary infections are adequately described, no discussion about this aspect of treatment is made. Therefore, the title of this case report is misleading.
Major remarks:
- Please discuss on the Carbapenem-resistant bacteria you identified during the treatment of your patient, especially on the following aspects: mechanisms of resistance (e.g. carbapenemases), possible antibiotic regimens, and outcome of infections with multi-resistant bacteria.
- The antibiotic regimen with meropenem was switched to an extended infusion over 3 hours (as recommend by some guidelines). Did you have any therapeutic drug monitoring? If yes, please provide information on drug levels. But please also discuss on aspects of therapeutic drug monitoring of antibiotics and implications on infusion time. Was the patient on dialysis? How could that affect antibiotic drug levels. Did you use a cytokine adsorber?
- I would also recommend discussing the perianal abscess in the context of being a possible initial focus of gram-negative bacteria.
- No details on the indication of ECMO therapy is provided: please provide information on pulmonary gas exchange and ventilator settings which led to installation of ECMO.
- The description of methods for microbiological processing and determination of antibiotic resistance is very detailed. I would recommend shortening this section and focusing only on relevant details which are needed for a common understanding of the case report. A detailed description may be provided as supplementary material.
Minor remarks:
- Table 1 is duplicated, please remove the duplicate
- In Figure 1: please correct perinatal abscess to perianal abscess
Reviewer 2 Report
The present case highlights the significant diagnostic and therapeutic
challenges of HAIs in patients with COVID-19 receiving long-term ECMO.
In fact, it is expressed how Covid's pathology and ventilation may predispose to the development of nosocomial infections by multi-resistant germs.
I agree that all patients with pathologies that lead to invasive ventilation with ventilators and for long periods of time, predispose to the development of over infection by organisms defined as "alerts".
Therefore, I do not believe that we should focus on the problem of Covid-19 because, in low-intensity wards, numerous studies have shown that the number of infections with multi-resistant micro-organisms has fallen considerably as a result of the hygiene and sterilization precautions taken during the Covid period.
I, therefore, recommend citing and reading the following recently published articles carefully if possible:
PMID: 33498701
PMID: 33031863
However, the article is well detailed and accurately presented in terms of images, captions, and English vocabulary.
Therefore, since it is not possible to state from a case report that covid-19 predisposes to the development of superinfections regardless of the assisted ventilation mechanisms, I propose to include the previously proposed articles in the limitations.
Good work.
Round 2
Reviewer 1 Report
Thank you for editing your manuscript accordingly. These improved the case report.
I only have two minor remarks:
1. Thank you for presenting the respiratory state of the patient which led to ECMO therapy. For me, PIP of 17 mbar is very confusing, especially when using a PEEP of 15 mbar. This would mean, that your driving pressure of ventilation was only 2 mbar - which is impossible. Or do you mean a Peak pressure (PIP) of 32 mbar with a PEEP of 15? Please clarify.
2. The respiratory state and ventilator settings of the second ECMO episode are exactly the same as the first episode. Please check if this is correct.
